



# Structure of Suasselkä Postglacial Fault in northern Finland obtained by analysis of local events and ambient seismic noise

Nikita Afonin[1], Elena Kozlovskaya[2,3], Ilmo Kukkonen[4] and DAFNE/FINLAND Working Group*

[1]Federal Centre for Integrated Arctic Research RAS, Arkhangelsk, Russia
[2]Oulu Mining School, POB-3000, FIN-90014, University of Oulu, Finland
[3]Geological Survey of Finland, P.O. Box 96, FI-02151, Espoo, Finland
[4]Department of Physics, University of Helsinki, P.O. Box 64, FI-00014, Helsinki, Finland
* A full list of authors and their affiliations appears at the end of the paper.

Correspondence to: Nikita Afonin (afoninnikita@inbox.ru)

**Abstract.** Understanding inner structure of seismogenic faults and their ability to reactivate is particularly important in investigating continental intraplate seismicity regime. In our study we address this problem using analysis of local seismic events and ambient seismic noise recorded by the temporary DAFNE array in northern Fennoscandian Shield. The main purpose of the DAFNE/FINLAND passive seismic array experiment was to characterize the present-day seismicity of the Suasselkä post-glacial fault (SPGF) that was proposed as one potential target for the DAFNE (Drilling Active Faults in Northern Europe) project. The DAFNE/FINLAND array comprised the area of about 20 to 100 km and consisted of 8 short-period and 4 broad-band 3-component autonomous seismic stations installed in the close vicinity of the fault area. The array recorded continuous seismic data during September, 2011-May, 2013. Recordings of the array have being analyzed in order to identify and locate natural earthquakes from the fault area and to discriminate them from the blasts in the Kittilä Gold Mine. As a result, we found several dozens of natural seismic events originating from the fault area, which proves that the fault is still seismically active. In order to study the inner structure of the SPGF we use cross-correlation of ambient seismic noise recorded by the array. Analysis of azimuthal distribution of noise sources demonstrated that during the time interval under consideration the distribution of noise sources is close to the uniform one. The continuous data were processed in several steps including single station data analysis, instrument response removal and time-domain stacking. The data were used to estimate empirical Green's functions between pairs of stations in the frequency band of 0.1-1 Hz and to calculate correspondent surface wave dispersion curves. The S-wave velocity models were obtained as a result of dispersion curves inversion. The results suggest that the area of the SPGF corresponds to a narrow region of low S-wave velocities surrounded by rocks with high S-wave velocities. We interpret this low velocity region as a non-healed mechanically weak fault damage zone (FDZ) that remained after the last major earthquake that occurred after the last glaciation.





## 1 Introduction

In studying of mechanisms of large earthquakes investigations of seismogenic fault structure and properties are of particular importance. One group of seismological studies concentrate on mapping the seismic source using recordings of seismic events. This includes mapping of the fault plane using distribution of hypocentres of earthquakes originating from the fault and also calculating orientation and dip of fault planes from seismograms of earthquakes (fault plane solution methods, centroid moment tensors). Another group of methods investigates inner structure of fault zones using structural geology, palaeoseismology, seismic reflection and refraction experiments and geodetic measurements. The studies of the second group show that inner structure of seismogenic faults is complex (Davis and Reynolds, 1996) and the main slip planes are surrounded by so-called fault damage zones (FDZ) which arise along the fault as a result of brittle deformation and cracking in rock in response to stress (Chester and Logan, 1986, Shipton and Cowie, 2003). These fault damage zones are mechanically weaker than surrounding rock and they can be detected as zones of low seismic velocities. Some recent investigations of FDZ produced by large earthquakes have demonstrated that the width of them can vary from several dozen meters to 1-2 km (Vidale and Li, 2003, Cochran et al., 2009). It is still debatable whether FDZ persists over a full earthquake cycle, which may last hundreds to thousands of years (Cochran et al., 2009) or the FDZ are healing during the years or decades following a main shock (Crone et al., 2003, Vidale and Li, 2003).

The question of longevity of faults is particularly important in investigating the continental intraplate seismicity regime (Stein, 2007). It is known that most such earthquakes can be related to detected fault zones, but continents contain many such features, of which only few are active. As suggested by Stein (2007), McKenna et al. (2007), long-lived, wide and mechanically weak damage zones concentrate intraplate strain release, hence hypocenters of future earthquakes would be located along these zones. However, if the damage zones are healed after the major shock, and are not significantly weaker than surrounding, then the intraplate seismicity would be a transient phenomenon that migrates among many fossil weak zones. That is why study of damage zones of faults which were not active during historical time may provide new information about intraplate seismicity phenomenon.

In our study we address the problem using an example of the Suasselkä post-glacial fault located in northern Fennoscandia (Fig. 1). Postglacial (PG) faults in northern Fennoscandia were formed during the last stages of the Weichselian glaciation (ca. 9,000 - 15,000 years B.P.), when reduced ice load and relaxation of accumulated tectonic stress resulted in rapid uplift in Fennoscandia and large-magnitude earthquakes with Mw of 7-8.2 (Wu et al., 1999, Olesen et al., 2004, Kukkonen et al., 2010). The length of the PGFs may vary from 2 to 150 km and the maximum height of the fault scarps from 1 to 12 m, yet in the extreme cases up to 30 m (see compilation in Olesen et al., 2004).

The Suasselkä Post-glacial fault (SPGF) of total length of 48 km is the longest PG-fault in Finland with strike of 35-50 deg and scarp of 0-3 m. It has been studied using magnetic and electromagnetic measurements by Paananen (1987) and Kuivamäki et al. (1998), who suggested that the fault dips southeast. The southeastern part of the fault is located in pre-existing fracture zone in the southeast, whereas existence of the oldest structure in the northwest remained uncertain. Based



on the radiocarbon dated buried organic materials the time frame of fault-activity ranges from 9730 to 5055 cal BP (Sutinen et al., 2014). The reprocessed seismic data along several reflection profiles crossing the fault area (Abdi et al., 2015) demonstrated complex structure of the fault area down to a depth of 2-3 km, with two sets of segmented and discontinuous dipping reflectors.

In our study we investigate the inner structure of the SPGF using distribution of hypocentres of local seismic events and analysis of ambient seismic noise recorded by the temporary DAFNE array. The ambient seismic noise analysis has been used recently by Hillers et al. (2014) in order to investigate the inner structure and properties of Calico Fault Zone in the Eastern California.

## 2 Data

The main objective of the DAFNE/FINLAND seismic passive experiment was to answer two major questions: a) whether the Suasselkä postglacial fault (SPGF) is still seismically active? b) If it is active, what is the geometry of its seismogenic zone and the depth to it? The project was initiated by several organizations in Finland (Geological Survey of Finland, Sodankylä Geophysical Observatory of the University of Oulu and Institute of Seismology of the University of Helsinki). The main purpose of the field experiment was to install an array of autonomous seismic stations in the target area of the

SPGF, in order to collect continuous seismic data for the period of September, 2011-May, 2013.

Selection of the DAFNE sites was done in August, 2011. Four sites with the permanent electric power supply were found in the vicinity of the target area (SPGF). The other 8 sites were selected taking into consideration surface position of the fault and the possible dip of the fault to the southeast. In the first part of September, 2011 the sites were prepared for installation and autonomous seismic recording instruments were installed to the sites during the second part of September, 2011.

Coordinates of all stations of the DAFNE array and description of the instrumentation used in the experiment is given in Table 1. The instruments were provided by the institute of Seismology of the University of Helsinki and by the Sodankylä Geophysical Observatory. Location of stations of the DAFNE array is shown in Fig. 1.

During the data acquisition period all the stations were regularly (once per two months at average) served by two staff members of the SGO for changing batteries and media and making the basic data quality control. Raw continuous data from

temporary stations of the DAFNE array collected during the service trips was copied to the data server of the SGO in their original formats. The main steps for data pre-processing included applying time corrections, conversion to the miniSEED format and merging the data files into miniSEED files of 24 hours length using IRIS PASSCAL software for processing and visualization of seismological data (http://www.iris.edu/manuals/SEEDManual_V2.4.pdf). Analysis of continuity of the data and estimation of noise level at the sites was also performed after each field trip. Generally, the noise level was higher

during daytime, and the seismograms recorded during the daytime were seriously contaminated by signals from production blasts in numerous mines in northern Sweden, Finland and Russia. Major problems were detected at stations DF10 and DF07, where the high noise level was due to technical problems with sensors and cables.



# 3 Detection and location of seismic events

At the beginning of the data analysis we tried to make a preliminary list of local seismic events using monthly seismic bulletins published by the Institute of Seismology of the University of Helsinki (FENCAT: http://www.seismo.helsinki.fi/english/bulletins/index.html). The criteria for events selection were coordinates of epicenters (from 67.50 N to 690 N and from 230 E to 280 E) and the timeframe corresponding to the DAFNE data acquisition period. The epicenters of these events are shown in Fig. 2. As seen, regional permanent seismic networks detected no natural events from the SPGF area. This can be partly explained by large distances between existing permanent seismic stations and the fault. The second problem for detection of natural seismic events in northern Finland is a huge number of production and development blasts originating from numerous mines and quarries. The DAFNE/FINLAND array recorded up to 100 of such blasts per day from northern Sweden, Russia and Finland. Due to this, it was not possible to use a routine LTA/STA analysis for automatic event detection. Therefore, the manual data analysis was used. The continuous data were accessed and reviewed with Seismic Handler Motif (SHM) program package (http://www.seismic-handler.org/portal). As a first step, we analyzed all the continuous data for the period of September, 2011 –October, 2011 and relocated all local events seen in the DAFNE data. This made it possible to compile a dataset of waveforms and time intervals of explosions from the mines and quarries outside the target area. This dataset was used to exclude such events from the further analysis. This was a necessary step, because about 30%‑50% of all such explosions are not included into the FENCAT bulletin and other regional bulletins. The continuous data was filtered by the Butterworth 3rd order 2-40 Hz bandpass filter and waveforms were analysed using SMH software. Totally, 1188 events in September, 2011-October, 2011 were analyzed and relocated using the DAFNE array data.

For relocation we used LocSAT seismic event location program (Bratt, 1988) and manually picked first arrivals of P- and S-waves. The first arrivals of P-waves were picked on Z component of the data filtered by the 3rd order Butterworth 2- 40 Hz bandpass filter. The first arrivals of S-waves were picked either on horizontal E or N component filtered by the Butterworth 3rd order 2-6 Hz bandpass filter, depending on the number of traces in which the same signal was better seen.

The distance between epicenter and stations of the array is less than 60 km for the events originating from the target area. In this case the first arrivals of P- and S-wave correspond to direct Pg and Sg waves refracting in the upper crust and the LocSAT procedure provides the hypocenter coordinates with the satisfactory precision for events detection.

In order to test the precision of the location procedure, we used blasts originating from the Kittilä Gold Mine. These blasts occurred inside known blasting time windows and have similar waveforms. Information about time windows for production and development blasts time was kindly provided by Engineering Superintendent of Agnico-Eagle Kittilä Mine André van Wageningen (personal e-mail communication).

The hypocenters of selected blasts obtained after relocation were concentrated inside the mining area that is about 5 km long and 2 km wide (www.agnicoeagle.com), with depths varying from 0 km to 1 km. This is a satisfactory location precision for





the LocSAT procedure and events detection. As a result of this preliminary analysis, we distinguished two types of events originating from our target area, but having different waveforms:

1) Blasts originating from the Kittilä mine;

2) Events originating from the SPGF area and its surrounding that could be of natural origin.

The waveforms of these events were used in further manual analysis of continuous data for the period till 31.05.2012. Totally, the DAFNE network recorded 10230 local events during September, 2011-May, 2012. From this amount we selected 40 events with the waveforms similar to those of the group 2) and located them using manually picked first arrivals of P- and S- waves and the LocSAT algorithm. Additional control on discriminating natural events from blasts in Kittilä Gold Mine was done using analysis of waveforms specra (Glitterman and van Eck, 1993, Glitterman et al., 1998).

Hypocentres of events identified as natural ones are presented in Table 2 and their epicentres are shown in Fig. 3. Precision of hypocentre coordinate determination using the LocSat was usually of an order of 2-5 km. In our study the depth of hypocenters of local earthquakes varies from about 2 to 15 km, with an average error of about 3-4 km. As seen, epicenters of many local earthquakes recorded by DAFNE/FINLAND array are spatially coincident with the Suasselkä post-glacial fault. This is indication that the fault is still seismically active.

**4 Ambient noise analysis**

Analysis of empirical Green's functions (EGFs) estimated from ambient seismic noise has been widely used in order to estimate seismic velocities in the subsurface (c.f. Shapiro and Campillo, 2004, Shapiro et al., 2005, Campillo, 2006). Poli et al. (2012, 2013a, 2013b) applied analysis of EGFs to retrieve body waves reflected from the Moho and upper mantle discontinuities and to estimate the 3D S-wave velocity of the upper crust of northern Finland. In our study we use the

procedure described in Poli et al. (2012, 2013b) in order to estimate EGFs from continuous recordings of vertical component of all stations of the DAFNE array. The functions are then used in order to estimate seismic velocities in the uppermost crust of the SPGF area.

**4.1 Analysis of spectrograms of the ambient noise**

Some stations of DAFNE temporary network were installed in the sites with high level of antropogenic noise (for example,

station DF02 was installed near the Kittilä Gold mine). Such a noise is not propagating to large distances from the noise source and this could cause distortion of EGFs. For identification of antropogenic noise on seismograms and selections of frequency band for filtering of the input data before cross-correlation, we used analysis of spectrograms estimated from continuous data recorded by stations closely located to a mine (DF01, DF02, DF09) and by the most distant station rom the mine (DF12). We studied high frequency and low frequency parts of spectrograms separately. Also, we studied the

spectrograms calculated from data recorded during weekend and workday. The examples are shown in Fig. 4 (a, b),





respectively. As seen, all the spectrograms are identical in the frequency band of 0.1-3 Hz and characteristics of the noise in this frequency ban do not depend of day of the week and stations location.

If the high frequencies are considered, we can see stable noise in the frequency band of ~13-14 Hz for workday (Fig. 5b). This noise was recorded by stations located closely to the mine (DF01, DF02, DF09) and it was not registered by the most

distant station from the mine (DF12). Therefore, it is antropogenic noise from the mine. It is worse to notice that station DF12 was installed near the river, and stable amplitude maximum at frequency of about 42 Hz is most probably noise of the river (Fig. 5b.).

Thus, we applied filtering in the band of 0.1-1 Hz for all input signals before computation of EGFs.

## 4.2 Analysis of azimuthal distribution of noise sources

Analysis of azimuthal distribution of ambient noise sources during the experiment data acquisition period is an important part of data preparation for EGFs calculation as the algorithm of their calculation functions is strongly dependent on directivity of noise sources. The algorithm is simpler if the noise sources have uniform azimuthal distribution. There are some methods of calculation of azimuthal distributions such as the f-k analysis (Neiddell and Taner, 1971, Douze and Laster, 1979) and beamforming (c.f. Schweizer et al., 2012). Configuration of the DAFNE array does not allow using f-k analysis,

unfortunately. That is why we applied beamforming procedure in the time domain to about 5000 surface wave parts of EGFs, calculated between station DF09 and every other station of DAFNE array. Green's functions were calculated from selected segments of continuous recordings bandpass filtered in the band of 0.1-1 Hz. Six segments per day with duration of one hour were selected for each day from 1.01.2012 to 30.04.2012 and from 10.12.2012 to 31.12.2012. During these time periods all the stations of the array were recording data. The result is shown in Fig. 6.

As one can see, during January and February, 2012 (days 0-50) the ambient noise was recorded from various directions, with prevailing azimuth of 0-200 degrees. Since 1.03.2012 the prevailing azimuths are of ~100 and 350-360 degrees and this tendency is slightly changing during March (days approximately 50-80) and April (days approximately 80-110). In December the prevailing azimuths are approximately 100-210 degrees. The observed changes in azimuthal distribution of noise sources may be caused by reduction of marine microseisms during winter months, because the range of 300-360

degrees corresponds to azimuths to the seashore in the North. This distribution is necessary to take into account when making interpretation of EGF.

## 4.3 H/V analysis of the ambient noise

The bedrock in our study area is covered by a thin (up to several dozens meters) layer of quaternary deposits developed during the late- Weichselian glaciation. In our study, we applied H/V analysis of ambient noise (Nakamura, 1989, SESAME

H/V User Guidelines, 2005) in order to estimate thickness of the sedimentary layer. It was necessary prior to calculation of velocity model using inversion of EGFs, because low velocities in this layer can affect inversion results.



We applied H/V analysis procedure implemented into Geopsy software (www.geopsy.org) to ambient seismic noise, recorded by selected DAFNE stations during winter and summer months. We used time segments selected from parts of recordings without antropogenic noise with duration of 30 minutes. Results obtained for winter and summer time were identical. Fig. 7 demonstrates an example of H/V analysis results for some of selected stations.

As seen from Fig. 7, the resonance frequency is between 20-40 Hz for all the data considered. From petrological data for sedimentary rocks (Gebrande et al., 1982, Kaikkonen, 2007) the S - wave velocities are about 250-350 m/s and P-wave velocities are about 500-700 m/s in our study area. Therefore, the maximum thickness of the sedimentary layer in our study area can be estimated as 5 m.

## 5 Calculating empirical Green's functions and dispersion curves

For evaluation of EGF we applied pre-processing procedure that includes pre-filtering in the frequency band of 0.1-1 Hz, deconvolution of instrument responses and removing record parts with earthquakes, quarry blasts and explosions and with high signal-to-noise ratio. After that we calculated cross-correlation functions between all pairs of the DAFNE stations. Whitening was not applied. Fig. 8 shows Green's functions calculated between station DF01 and all other stations of the DAFNE array.

In Fig. 8 we can see that EGFs are asymmetric for some pairs of stations. This effect is not caused by propagation direction of ambient noise because during the experiment the noise was recorded from all azimuths for all azimuths (Fig. 6). The asymmetry can be explained by different distances between stations and selected frequency band. In Fig. 8 we can see that EGFs functions are asymmetric for interstation distances less than 14 km and that after this distance the EGFs are symmetric. In the frequency band of 0.1-1 Hz the wave length is 3-30 km for phase velocity of about 3000 m/s. That is why for

interstation distances less than 15 km the casual and acasual parts of EGFs are overlapping. Some of the EGFs have different amplitude of casual and acasual parts also for interstation distances more than 15 km. This effect is caused by dominant propagation direction of ambient noise from some azimuths. But in our study we use phase characteristics of EGSs (time of propagation); that is why different amplitudes of casual and acasual parts is not the problem.

For extracting of dispersion curves, we applied narrow bandpass filters with widths of 0.125 Hz to surface wave parts of all

EGFs. Some of the station pairs of the DAFNE array have interstation distances less than wavelength for some frequencies; that is why calculated velocities were with large error bars. These velocities were excluded from further processing. Some dispersion curves contain fundamental and the first higher mode of Rayleigh wave (Fig. 9). In our study we did not use the first higher mode and used fundamental mode only.

As the scatter of dispersion curves was large, we calculated averaged dispersion curve from all EGFs. We also calculated

separately averaged dispersion curve for EGFs corresponding to two groups of pairs of stations. The first group (referred hereafter as Group 1) is composed of the pairs in which stations are installed on different sides of the fault and the second





group (hereafter Group 2) is composed of the pairs of stations installed on the same side of the fault. As Fig. 10 shows comparison of two different dispersion curves corresponding to Group 1 and Group 2.

In Fig. 10 one can see that dispersion curve for Group 1 indicate significantly smaller velocities than the dispersion curve for Group 2. This result suggests that seismic velocities inside the fault zone are most probably significantly different from those

outside the fault zone. Therefore, we decided to determine this difference using inversion of these two dispersion curves separately.

## 6 Inversion of dispersion curves

For inversion of dispersion curves we used the Geopsy software ([www.geopsy.org](http://www.geopsy.org)). The software uses Neighborhood global optimisation algorithm by Sambridge (1999) modified by Wathelet (2008). As solutions of geophysical inverse problems are

generally non-unique, the ideal solution in this method is made of the ensemble of all models that equally fits the data and prior information.

The starting model we used in inversion is presented in Table 3. Parameters of the uppermost layer of the model (Vp, Vs and density) were obtained from results of H/V analysis presented in our study and parameters for the other layers were constrained using the velocity models from previous study by Janik et al. (2009). The models by Janik et al. (2009) were

calculated for seismic controlled-source seismic profiles closest to our study area. We applied about 500 iterations for calculation of 1D velocity models with minimum misfit function value. The results of inversion for two dispersion curves corresponding to Group 1 and Group 2 are presented in Fig. 11(a,b) and Fig. 12(a,b). Fig. 11(a,b) shows two ensembles of calculated dispersion curves compared to two observed dispersion curves and Fig. 12(a,b) shows correspondent two ensembles of velocity models. The values of misfit function for each element of solutions ensemble are denoted by colour

scale in both figures. The error bars for dispersion curve for pair of stations outside the fault are larger than those for dispersion curve for pair of stations across the fault. This can be explained by variations of seismic velocities in the near fault zone.

Fig. 12(a) shows that 1D S-wave velocity model estimated from dispersion curves for pairs of stations of Group 1 is consisting of 3 layers with low velocities (about 900 – 1400 m/s). The pronounced velocity boundary is located at a depth of

about 1000 m. For the pairs of stations of Group 2 the 1D velocity model consists of 2 layers with generally higher than in the previous case velocities varying from 2500 to 3500 m/s. As the Green functions estimated by cross-correlation of recordings of two stations are depending on structure between them, this result suggest that there exist an area inside the fault zone with seismic velocities significantly lower than those of bedrock outside the fault zone.

## 7 Discussion and conclusions

Two major results obtained in our study can be formulated as follows:



1) Suasselkä Post-Glacial Fault is still seismically active, as shown by distribution of hypocentres of local earthquakes from the fault area detected by the DAFNE array;

2) Analysis and inversion of averaged dispersion curves obtained from EGFs for two groups of seismic stations pairs (e.g. the pairs in which stations are located on opposite sides of the fault and the pairs in which stations are located outside the fault) revealed significant (about 1000 m/s) difference in seismic velocities inside the fault zone and outside the fault zone.

As shown in Fig. 1 and Fig. 2, regional seismic networks did not register any earthquakes originating from the SPGF area neither prior to DAFNE experiment no during the time period considered in Section 3. In spite of that, a number of events were revealed using the data of dense DAFNE array installed in the vicinity of the fault. This suggests that the magnitudes of these events are smaller than the detection magnitude threshold for the present configuration of seismic stations. Analysis of the FENCAT catalogue shows that no events originated from northern Finland with local magnitude ($M_L$) less than 0.7 were detected during DAFNE data acquisition period. Hence our study suggests that local magnitudes of events detected in our study are less than 0.7, while the value of 0.5 can be assumed as a conservative estimate. The more precise magnitude estimate was difficult to do in our study as the events were seen on filtered recording only and band-pass recordings distort the recording amplitudes.

Concerning the low velocities inside the fault area, these velocities cannot be explained by sediments, because results of the H/V analysis presented in Section 4.3 suggest that the thickness of sediments is no more than 5 m in our study area. That is why the most plausible explanation is that the low velocity area is a fractured zone inside the fault that can correspond to the fault damage zone (FDZ). The velocities outside the fault zone are typical S-wave velocities in the felsic rocks of the uppermost crystalline crust documented in pertophysical study of Finnish bedrock by Kern et al. (1993). The rough estimate of the width of this zone corresponds to a smaller wavelength in the considered frequency band of 0.1-1 Hz, that is about 1.5 km.

The velocity boundary at a depth of about 1200 km is seen in both velocity models obtained from average dispersion curves for Group 1 and Group 2. Origin of this boundary is not clear, but it cannot correspond to the lower boundary of the FDZ as seismic velocities obtained for the Group 1 are lower than those for the Group 2 down to the depth of 1300 m. However, we can speculate that the lower boundary of the fault damage zone may correspond to the sub-horizontal area of high reflectivity revealed by Abdi et al. (2015). The authors also demonstrated complex structure of the fault area down to a depth of 2-3 km, with two sets of segmented and discontinuous dipping reflectors. Slip on small-scale fractures corresponding to these reflectors may be the cause of small magnitude events detected by our study.

In summary, our study revealed that the SPGF is still seismically active and there exist non-healed mechanically weak fault damage zone (FDZ) in the fault area that remained after the last major earthquake that occurred after the last glaciation. This suggests that SPGF has the potential for future reactivation. Our study also confirms that analysis of EGFs estimated by cross-correlation can be an efficient tool in investigating inner structure of seismogenic faults.

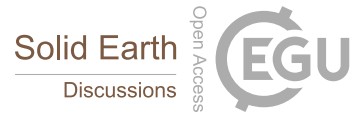

## 8 Acknowledgements

The present paper was a part of research projects DAFNE (Drilling into Active Faults in Northern Europe). The DAFNE/FINLAND Working Group consists of the following individuals: Ilmo Kukkonen (University of Helsinki, Department of Physics), Pekka Heikkinen (University of Helsinki, Institute of Seismology), Kari Komminaho (University of

Helsinki, Institute of Seismology), Elena Kozlovskaya (Oulu Mining School, University of Oulu/ Geological Survey of Finland), Riitta Hurskainen (Sodankylä Geophysical Observatory, University of Oulu), Tero Raita (Sodankylä Geophysical Observatory, University of Oulu), Hanna Silvennoinen (Sodankylä Geophysical Observatory, University of Oulu).

The study was partly funded by Posiva Oy and Geological Survey of Finland. The authors are thankful to the technical unit of the SGO for service of the DAFNE station during experiment. Our particular thanks are to Mrs. Inna Usoskina, who

performed a tremendous work with manual analysis of the FINLAND/DAFNE data and events relocation for this study. The digital aeromagnetic map of Finland was provided by the Geological Survey of Finland. Geopsy software (www.geopsy.org) was used for ambient noise analysis and inversion of dispersion curves.

This work was supported in part by the Federal Agency of Scientific Organizations, project № AAAA-A16-116052710111-2

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



Table 1. Information about seismic stations of the DAFNE/FINLAND temporary array

| Name | Location | Coordinates | | | Operation started | | | | | Group | Sensor type | Logger | Sampling rate (sps) |
|------|----------|-----|------|---------------|-----|-------|------|------|-----|------|-------------|--------|---------------------|
| | | Lat | Long | height (m) | Day | Month | Year | Hour | Min | | | | |
| DF01 | Rautuskylä | 67.8758 | 25.0514 | 241 | 13 | 9 | 2011 | 7 | 17 | Oulu | Trillium compact | PR6-24 | 100 |
| DF02 | Lehto | 67.8566 | 25.3862 | 227 | 23 | 9 | 2011 | 13 | 50 | Oulu | Trillium compact | Reftek130 | 100 |
| DF03 | Kapsajoki | 67.9403 | 25.1931 | 244 | 16 | 9 | 2011 | 14 | 7 | Helsinki+Oulu | Lennartz 3Dlite | Reftek130 | 100 |
| DF04 | Kiimalaki | 67.9886 | 25.1042 | 274 | 16 | 9 | 2011 | 15 | 22 | Oulu | Mark L4a | Reftek130 | 100 |
| DF05 | Outa-Perttunen | 68.0715 | 25.4131 | 302 | 15 | 9 | 2011 | 12 | 30 | Oulu | Mark L4a | Reftek130 | 100 |
| DF06 | Mietrikkilehto | 68.1015 | 25.6388 | 314 | 15 | 9 | 2011 | 9 | 47 | Helsinki | Lennartz 3Dlite | Reftek130 | 100 |
| DF07 | Suasselkä | 68.0262 | 25.4365 | 286 | 16 | 9 | 2011 | 6 | 34 | Helsinki | Lennartz 3Dlite | Reftek130 | 100 |
| DF08 | Arabiankangas | 68.0395 | 25.6588 | 326 | 15 | 9 | 2011 | 7 | 51 | Helsinki | Lennartz 3Dlite | Reftek 130 | 100 |
| DF09 | Salo | 67.9091 | 25.0926 | 227 | 14 | 9 | 2011 | 7 | 10 | Oulu | Trillium compact | PR6-24 | 100 |
| DF10 | Tepsanjänkkä | 67.9595 | 25.5397 | 271 | 15 | 9 | 2011 | 14 | 40 | Helsinki | Lennartz 3Dlite | Reftek130 | 100 |
| DF11 | Rajalompolontie | 68.0110 | 25.7622 | 340 | 17 | 9 | 2011 | 7 | 21 | Helsinki | Lennartz 3Dlite | Reftek130 | 100 |
| DF12 | Pokka | 68.1596 | 25.7760 | 282 | 13 | 9 | 2011 | 14 | 36 | Oulu | Trillium compact | SeisComP | 100 |



Table 2. List of seismic events in the area of Suaselkä Postglacial Fault detected by DAFNE seismic array

| Year | Month | Day | Hour | Min | Sec | Lat (deg) | Long (deg) | Depth (km) |
|------|-------|-----|------|-----|-----|-----------|------------|------------|
| 2011 | 9 | 28 | 22 | 8 | 59 | 67,85 | 24,86 | 12,5 |
| 2011 | 10 | 8 | 22 | 26 | 13 | 67,31 | 24,67 | 5,2 |
| 2011 | 11 | 17 | 1 | 18 | 10 | 68,01 | 25,50 | 0,0 |
| 2011 | 11 | 17 | 1 | 18 | 10 | 68,01 | 25,51 | 4,3 |
| 2011 | 12 | 3 | 18 | 5 | 52 | 68,45 | 23,41 | 1,0 |
| 2011 | 12 | 7 | 22 | 36 | 16 | 67,94 | 25,51 | 6,3 |
| 2011 | 12 | 9 | 3 | 47 | 51 | 68,47 | 25,92 | 7,5 |
| 2011 | 12 | 10 | 5 | 47 | 18 | 68,26 | 23,60 | 3,1 |
| 2011 | 12 | 12 | 12 | 50 | 23 | 67,85 | 25,21 | 5,2 |
| 2011 | 12 | 24 | 4 | 52 | 44 | 67,25 | 25,59 | 4,9 |
| 2011 | 12 | 24 | 19 | 42 | 1 | 68,44 | 25,82 | 7,5 |
| 2011 | 12 | 31 | 3 | 24 | 27 | 67,79 | 24,58 | 0,0 |
| 2012 | 1 | 7 | 7 | 25 | 40 | 67,79 | 25,07 | 6,2 |
| 2012 | 1 | 12 | 0 | 15 | 19 | 67,55 | 24,12 | 8,6 |
| 2012 | 1 | 17 | 3 | 22 | 39 | 68,12 | 24,21 | 17,0 |
| 2012 | 1 | 18 | 21 | 44 | 15 | 68,14 | 25,93 | 6,5 |
| 2012 | 1 | 20 | 5 | 36 | 31 | 68,14 | 25,90 | 7,3 |
| 2012 | 1 | 23 | 16 | 36 | 47 | 67,10 | 25,72 | 6,5 |
| 2012 | 1 | 23 | 23 | 24 | 23 | 67,81 | 25,08 | 6,8 |
| 2012 | 2 | 11 | 18 | 44 | 4 | 67,99 | 25,57 | 0,0 |
| 2012 | 2 | 18 | 0 | 33 | 4 | 68,14 | 25,93 | 0,0 |
| 2012 | 2 | 19 | 1 | 24 | 20 | 67,77 | 24,68 | 8,5 |
| 2012 | 3 | 4 | 14 | 22 | 44 | 67,92 | 25,40 | 0,0 |
| 2012 | 3 | 4 | 23 | 27 | 31 | 67,92 | 25,40 | 0,0 |
| 2012 | 3 | 19 | 23 | 49 | 17 | 68,23 | 23,93 | 2,5 |
| 2012 | 4 | 9 | 1 | 3 | 41 | 68,19 | 24,15 | 14,2 |
| 2012 | 4 | 15 | 15 | 39 | 29 | 67,47 | 24,46 | 4,6 |
| 2012 | 4 | 17 | 16 | 0 | 19 | 68,19 | 26,14 | 1,3 |
| 2012 | 4 | 18 | 20 | 47 | 32 | 67,71 | 27,43 | 9,5 |
| 2012 | 4 | 20 | 11 | 28 | 9 | 68,09 | 25,75 | 0,2 |
| 2012 | 4 | 22 | 7 | 25 | 41 | 67,80 | 24,77 | 5,8 |
| 2012 | 4 | 24 | 8 | 58 | 5 | 68,16 | 23,19 | 7,8 |
| 2012 | 4 | 26 | 16 | 38 | 51 | 67,68 | 24,01 | 2,4 |
| 2012 | 4 | 30 | 4 | 15 | 1 | 68,95 | 23,57 | 0,0 |
| 2012 | 5 | 1 | 9 | 29 | 39 | 67,91 | 25,40 | 0,0 |
| 2012 | 5 | 8 | 22 | 38 | 34 | 67,44 | 24,27 | 11,1 |
| 2012 | 5 | 14 | 22 | 10 | 11 | 67,28 | 25,80 | 7,1 |
| 2012 | 5 | 16 | 10 | 55 | 31 | 67,39 | 25,92 | 0,0 |
| 2012 | 5 | 21 | 14 | 16 | 24 | 68,50 | 25,78 | 2,1 |
| 2012 | 5 | 25 | 18 | 27 | 58 | 67,67 | 27,68 | 4,7 |



Table 3. Parameters of the starting velocity model for inversion of dispersion curves.

| Depth (m) | Vp (m/s) | Vs (m/s) | Density (kg/m$^3$) |
|---|---|---|---|
| 5 | 700 | 350 | 1500 |
| 500 | 6150 | 3590 | 2030.27 |
| 1300 | 6350 | 3710 | 2096.27 |
| 18000 | 6300 | 3650 | 2079.77 |





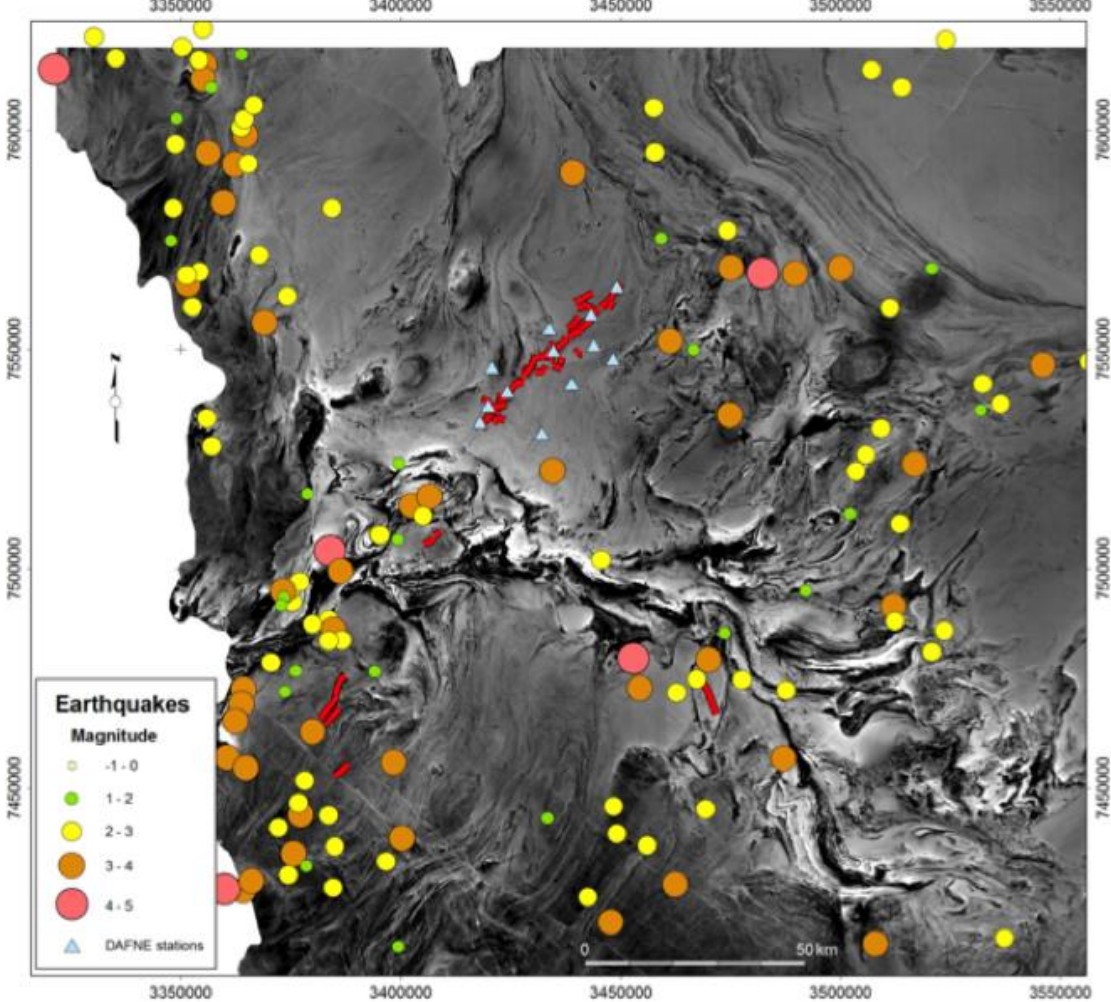

Figure 1. Position of DAFNE temporary array (blue triangles) on the aeromagnetic field map of Finland. Epicenters of local earthquakes detected by regional seismic networks in Fennoscandia prior to DAFNE/FINLAND experiment are shown by circles with size proportional to the magnitude of the event. Post-glacial faults in the area are shown by red lines. The coordinate system is Finnish National Coordinate System (KKJ).



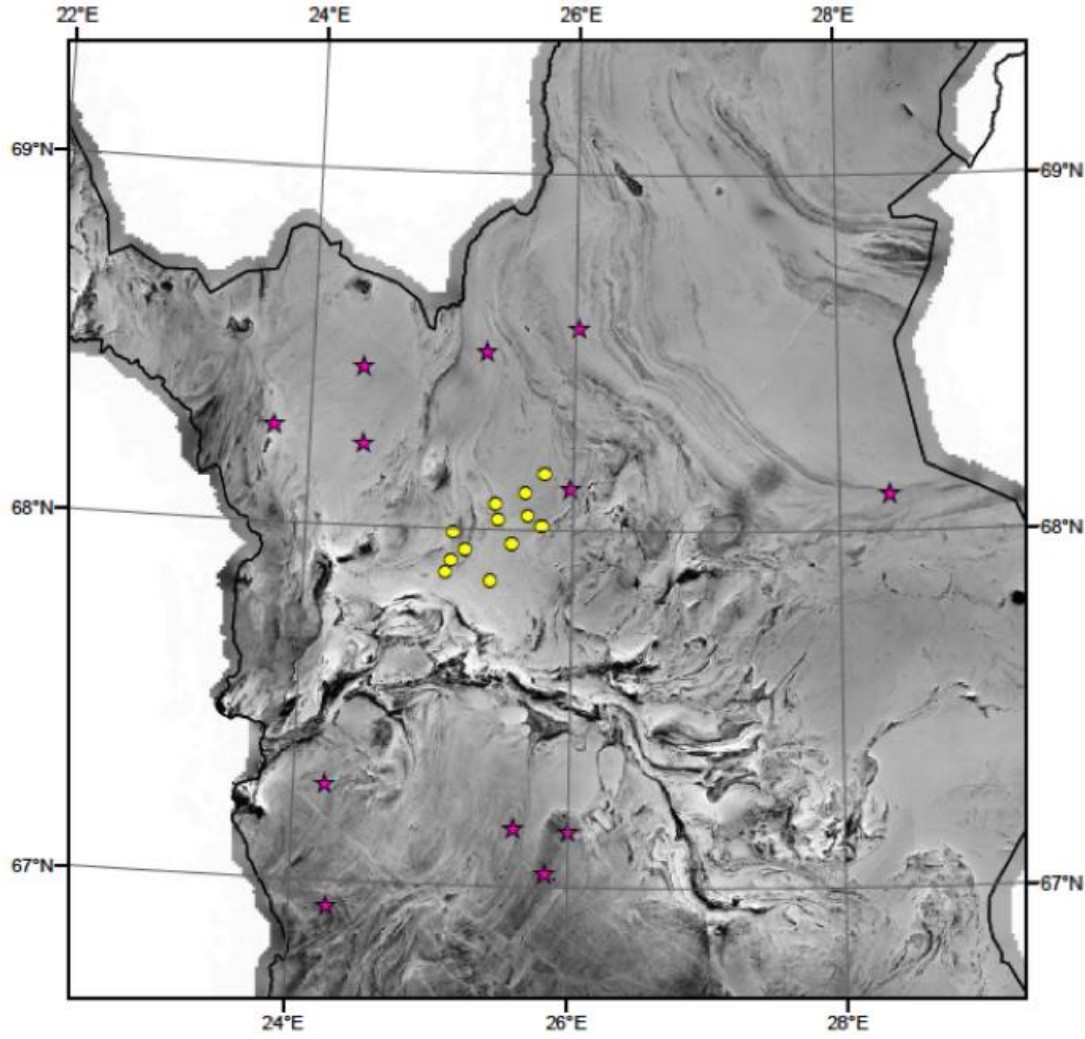

Figure 2. Position of DAFNE temporary array on the aeromagnetic field map of Finland (yellow dots). Epicentres of local earthquakes detected by regional seismic networks in Fennoscandia during September, 2011-May, 2012 are shown with magenta stars.





Figure 3. Epicentres of local earthquakes (stars) recorded by the DAFNE/FINLAND temporary during September, 2011-
May, 2012 shown on the aeromagnetic map of Finland.




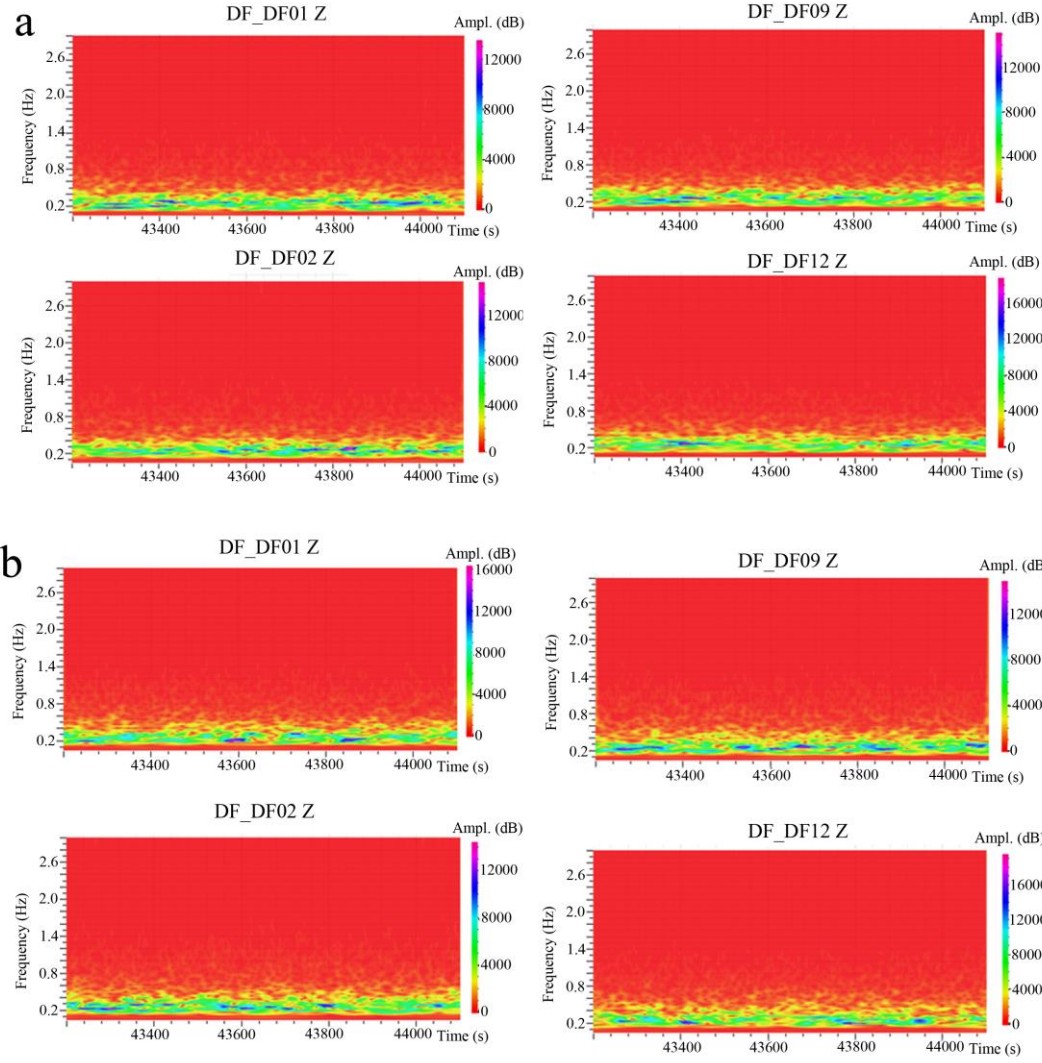

Figure 4. Examples of typical ambient noise spectrograms estimated for low frequency band (0.1-3 Hz) a) during weekend; b) during workday.

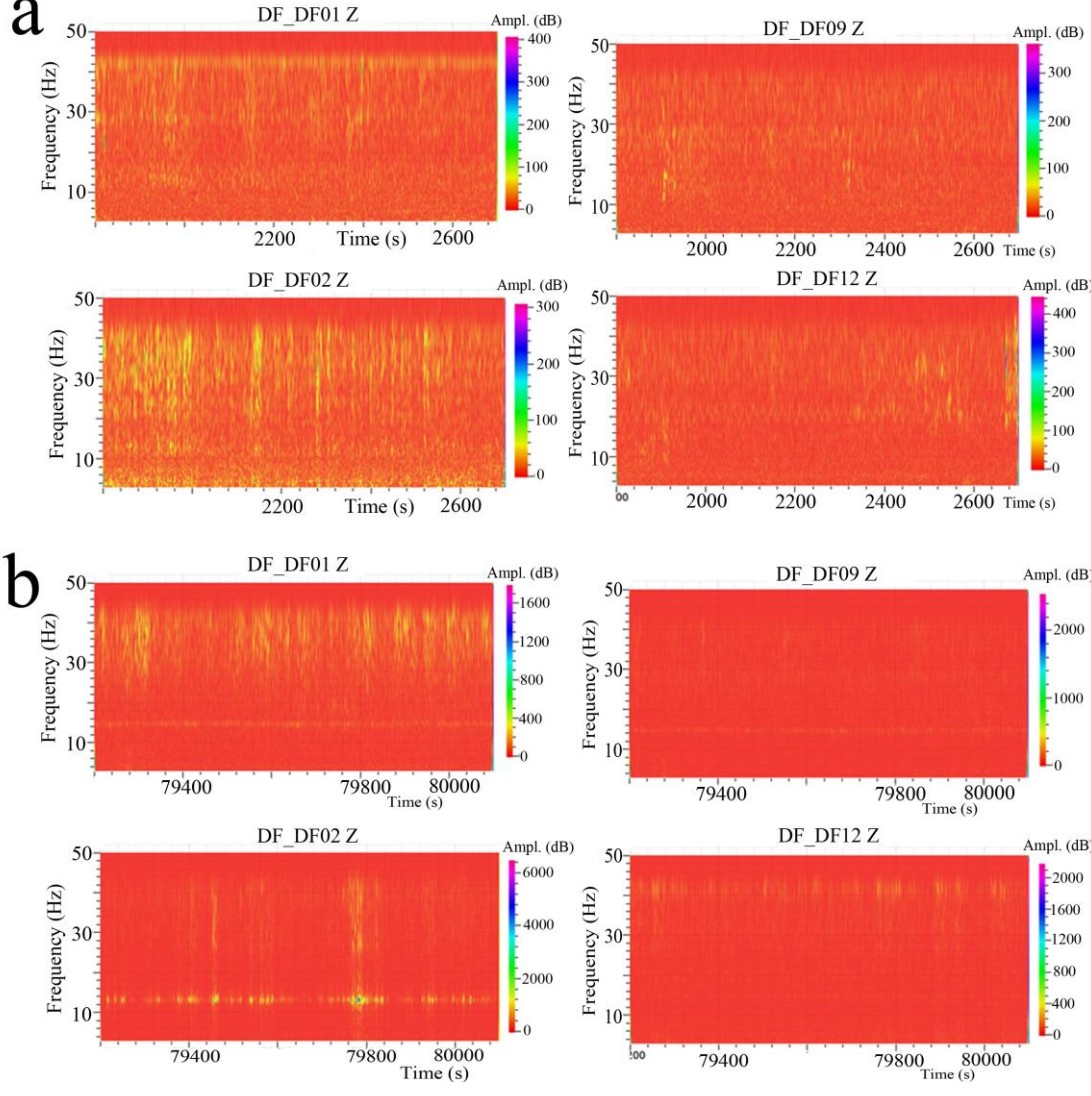

Figure 5. Examples of typical ambient noise spectrograms estimated for high frequency band (3-50 Hz). a) during weekend;
b) during workday;



## Azimuthal distribution of sources

Figure 6. Azimuthal distribution of ambient noise sources during time periods from 1.01.2012 to 30.04.2012 and from 10.12.2012 to 31.12.2012 estimated by beamforming in the time domain.

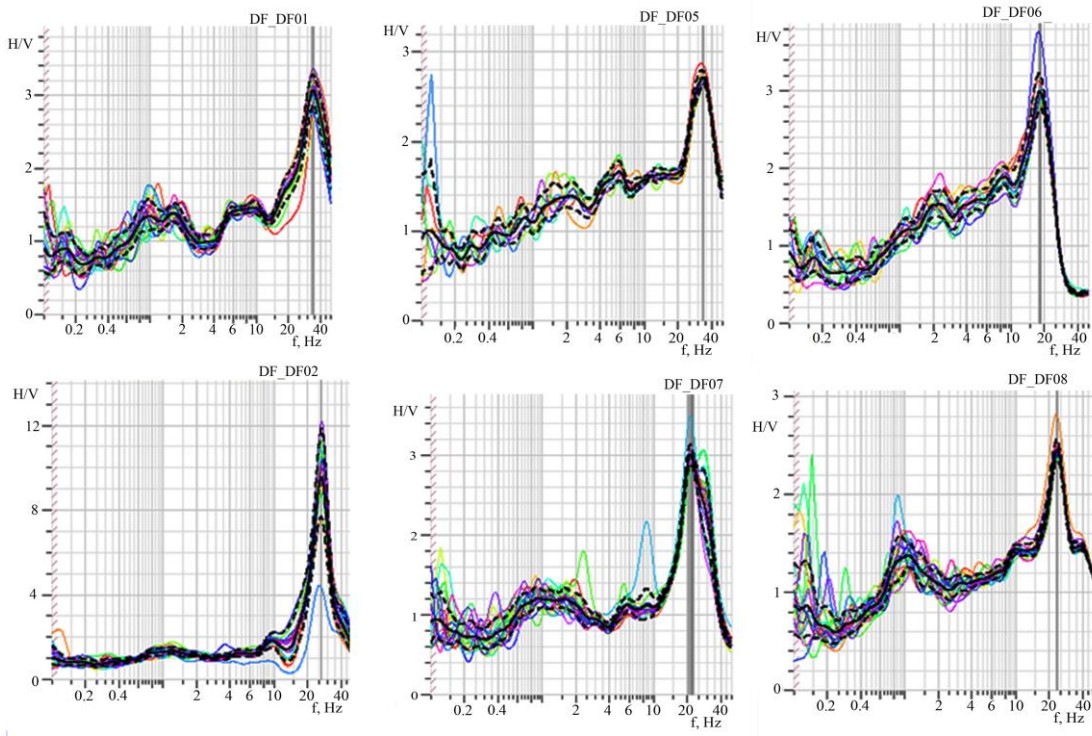

Figure 7. Results of H/V analysis for selected stations of the DAFNE array. The resonance frequency is marked by grey bar with width corresponding to error bar.



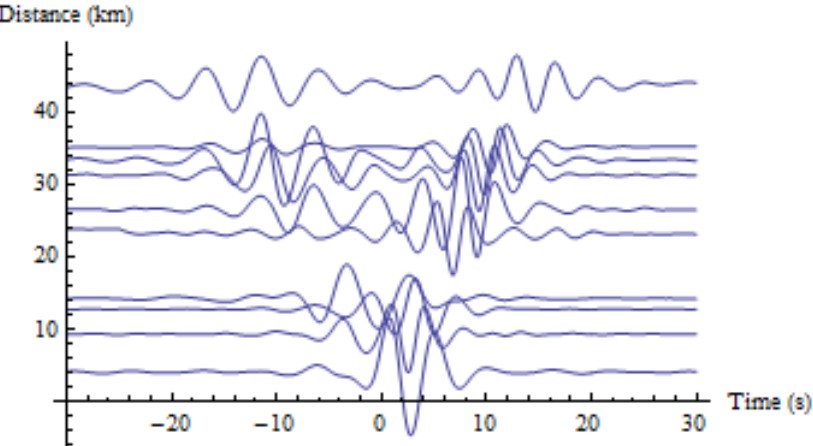

Figure 8. Empirical Green's functions calculated between station DF01 and all other stations of the DAFNE array.

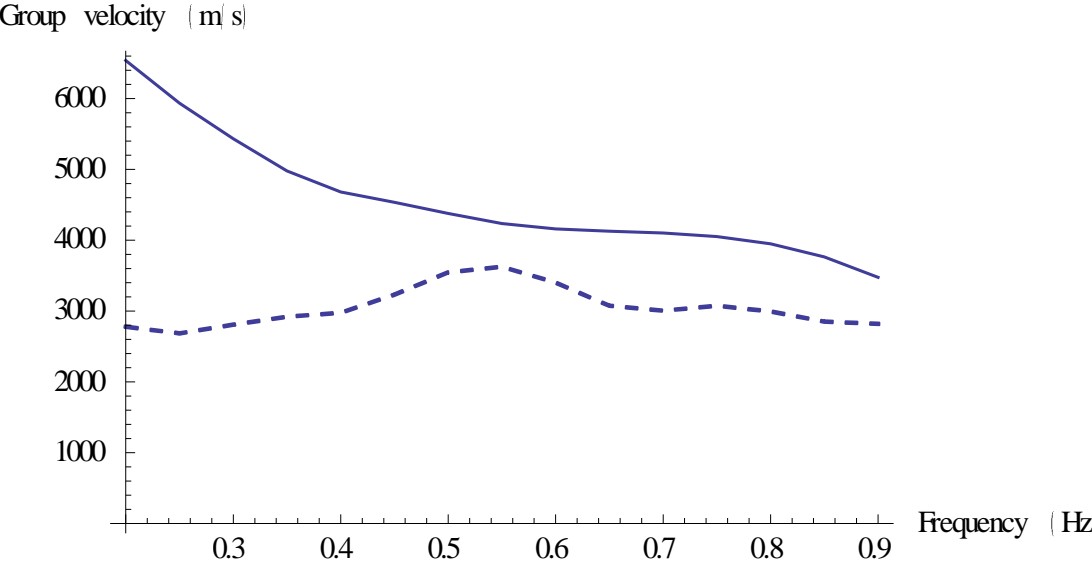

Figure 9. Example of dispersion curve with fundamental and the first order higher mode of Rayleigh wave estimated from EGFs. The dashed line denotes the  fundamental mode and the solid line denotes the first higher mode.



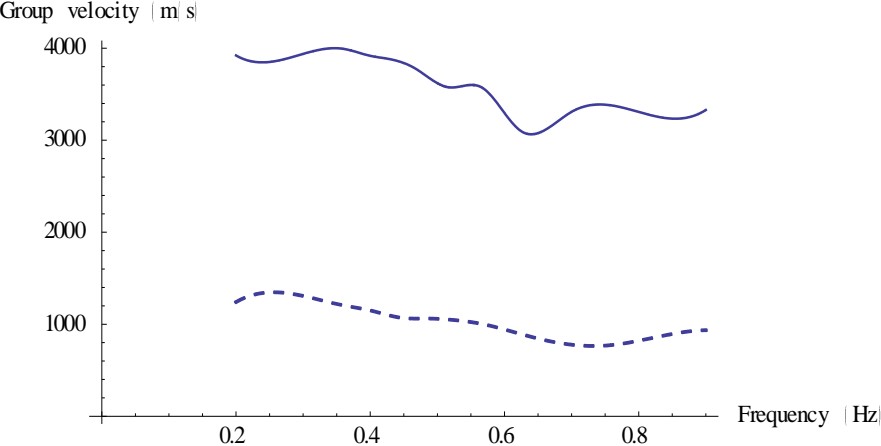

Figure 10. Averaged dispersion curves calculated from empirical Green's functions estimated for two groups of seismic stations pairs (see explanation for the Group 1 and Group 2 in the text). Dashed line denotes averaged dispersion curve for the Group 1 and the solid line denotes averaged dispersion curve for the Group 2.

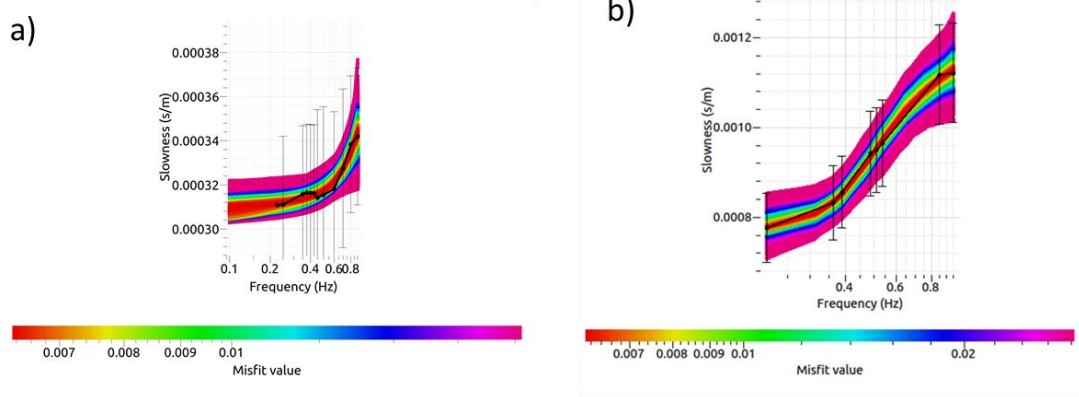

Figure 11. Results of inversion of averaged dispersion curves for station pairs of Group 1 (a) and Group 2 (b). The plots demonstrate the fit of model dispersion curves to the observed dispersion curve marked by solid black line with the corresponding error bar. Ensemble of all models that equally fits the data and prior information is shown by color plots in which the color scale indicates the value of misfit function.





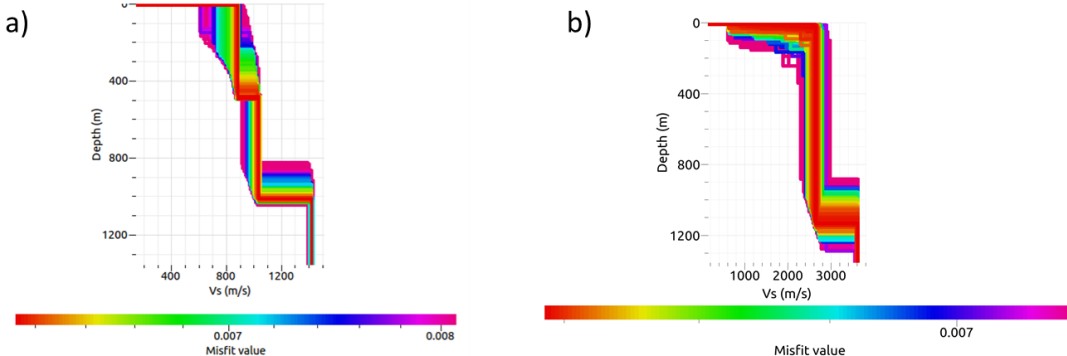

Figure 12. Results of inversion of averaged dispersion curves for station pairs of Group 1 (a) and Group 2 (b). The plots demonstrate ensemble of all velocity models that equally fit the data and prior information. The color scale indicates the value of misfit function.