# Peer review of "Structure of Suasselkä Postglacial Fault in northern Finland obtained by analysis of local events and ambient seismic noise"

_Solid Earth, 2016_

## Short Comment (SC1) · 22 Jul 2016

Dear authors: I went quickly through this discussion article and would like to raise a few minor comments:

You write: "The length of the PGFs may vary from 2 to 150 km and the maximum height of the fault scarps from 1 to 12 m, yet in the extreme cases up to 30 m (see compilation in Olesen et al., 2004)."

The paper by Olesen et al. (2004) likely does not include the most recent PGFs reported by Mikko et al.; Smith et al. and Malehmir et al. from central part of Sweden among others and this could be included and mentioned that these faults are no longer

confined to only northern parts of the northern countries as previously thought to. However, they may not be active like others in the north and in your case.

You write: "That is why the most plausible explanation is that the low velocity area is a fractured zone inside the fault that can correspond to the fault damage zone (FDZ)."

Multi-phase deformation zones are likely responsible for the location of most of PGFs. See the recent study by Malehmir et al. (the same journal) and how they conclude an existing and earlier structure was responsible for the Bollnäs PGF and the complexity of the situation. A major low-velocity zone was also observed there to an extent that delineation of a fresh bedrock movement was impossible to observe (highly fractured and crushed rocks).

There are also typos in the text here is an example I spotted: The velocity boundary at a depth of about 1200 km is seen in both velocity models obtained from average dispersion curves for Group 1 and Group 2.

I guess this meant 1200 m!

Suggested refs: Brandes, C., Winsemann, J., Roskosch, J., Meinsen, J., Tanner, D.C., Frechen, M., Steffen, H., and Wu, P.: Activity of the Osning thrust during the late Weichselian: Ice-sheet and lithosphere interactions, Quaterny Sci. Rev., 38, 49–62, doi:10.1016/j.quascirev.2012.01.021, 2012.

Malehmir, A., Andersson, M., Mehta, S., Brodic, B., Munier, R., Place, J., Maries, G., Smith, C., Kamm, J., Bastani, M., Mikko, H., and Lund, B., 2016. Post-glacial reactivation of the Bollnäs fault, central Sweden - a multidisciplinary geophysical investigation. Solid Earth, 7, 509–527.

Mikko, H., Smith, C. A, Lund, B., Ask, M., and Munier, R.: LiDAR-derived inventory of 25 post-glacial fault scarps in Sweden, J. Geol. Soc. Sweden, 137, 334–338, doi:10.1080/11035897.2015.1036360, 2015.

Smith, C., Sundh, M., and Mikko, H.: Surficial geologic evidence for early Holocene

faulting and seismicity, Int. J. Earth Sci., 103, 1711–1724, 2014.

Best regards, Alireza Malehmir

---

## Referee Comment (RC1) · G. Hillers (Referee) · 1 Aug 2016

Dear Editor,

Please consider my comments to the manuscript entitled

Structure of Suasselka Postglacial Fault in northern Finland obtained by analysis of local events and ambient seismic noise

by Afonin et al. (doi 10.5194/se-2016-90).

The authors analyse data from a temporary network consisting of 12 stations deployed in the vicinity of the Suasselka post-glacial fault to study the fault inner structure.

Many analysis tools are used, which support a complementary approach to fault zone characterization. However, I feel that parts of the analysis and parts of the presentation and discussion can be improved. The manuscript should be suitable for publication in Solid Earth Discussions after some moderate to major revision.

Please also note the supplement to this comment:
http://www.solid-earth-discuss.net/se-2016-90/se-2016-90-RC1-supplement.pdf

———————————————

[Figure]

**Supplement:**

Dear Editor,

Please consider my comments to the manuscript entitled

Structure of Suasselka Postglacial Fault in northern Finland
obtained by analysis of local events and ambient seismic noise

by Afonin et al. (doi 10.5194/se-2016-90).

The authors analyse data from a temporary network consisting of 12 stations deployed in the vicinity of the Suasselka post-glacial fault to study the fault inner structure. Many analysis tools are used, which support a complementary approach to fault zone characterization. However, I feel that parts of the analysis and parts of the presentation and discussion can be improved. The manuscript should be suitable for publication in *Solid Earth Discussions* after some moderate to major revision.

I waive my anonymity,

Gregor Hillers

General comments:

The authors discuss aspects of low-velocity damage zones and some resulting implications for faulting. However, in the introduction, they refer only to damage zones in strike-slip faulting environments. These damage zones are persistent features of mature strike-slip fault zones. They evolve over geologic times; their width and the velocity reduction depends roughly on the cumulative fault offset. In addition to this persistent structure, the degree of the velocity reduction, being a proxy for the material's degree of granularity, varies on time scales associated with the seismic cycle. How does this concept relate to the context of post-glacial faults discussed here? What is the cumulative offset of such faults in general and of the Suasselka fault in particular? What is the sense of motion? Under these circumstances, is the signature of a distinct low-velocity zone expected to be resolved?

Considering the earthquake analysis, the database of (downdip) earthquakes observed with the network could be further considered for reconstruction of observables that can be used to infer fault structure. (The authors claim that 2-15 km deep seismicity indicates that the fault is still active (line 14 page 5). Does the fault reach 15 km deep?) Fault zone head or trapped waves can provide important, high-resolution information on material contrasts along and within fault zones. Distinguishing events that do or do not excite head or trapped waves indicate dis-/continuous structural elements.

Considering the noise analysis, the authors group cross-fault correlation pairs (termed Group 1) and pairs of stations located on either side of the fault (Group 2). (How are pairs including the 6 stations (50% of the stations) located on top of the mapped fault (Fig. 1) classified?) This approach may certainly be motivated by the relatively sparse station spacing. However, it also

reflects the first-order concept of a low-velocity zone sandwiched betweent two competent blocks. I think the authors must discuss the application of this model better.

Then, the width of the inferred low-velocity region from group-1 results is simply estimated to correspond to the smaller wavelength in the considered frequency band, 1.5 km. This is quite a large value for a fault characterized by a small cumulative offset (the authors may compare this to values obtained from the normal fault hosting the L'Aquila earthquake—or any other well-studied fault exhibiting similar characteristics, but not strike slip faults in California). What prevents the authors from regionalizing the results as in doi:10.1007/s00024-014-0872-1? First, dispersion curves can be obtained from each pair. This yields, second, 2-D lateral group velocity maps on a grid, which can, third, be inverted for local shear wave profiles.

I think the figures can very much be improved. Is the data from the "areomagnetic" map used as a background synonymous with topography? If not, why is (areo)magnetic data used in the background? Figures 1-3 can be merged into two figures, perhaps even into just one. Please provide a large-scale inset that shows the target region, say, in relation to Scandinavia or Finland. The red "lines" under the DAFNE network in Figure 1 appear as rectangle, boxes. Show the seismicity in Figure 3 in relation to the mapped Suasselka fault. Consider an inset that shows the depth distribution of the seismicity (with error bars).

Figure 4: Why not use white for zero amplitude.The figures all look the same; that is what the authors point out in the text, but I wonder whether elimination of redundant data wouldn't be a better strategy for visualization here. I find the colorrange not exhausted in Figure 5.

Specific comments:

page 1, lines 11, 17: I doubt whether the network with station spacings on the order of 10 km allows probing of the "inner structure" of the fault, except, as said, if high-resolution approaches based on fault zone trapped and head waves are considered.

p 1, l 28: More than one rupture is needed to create a low-velocity zone (see above).

P 2, l 15: Consider references to earthquake and noise tomographies in fault zone environments by A. A. Allam et al. and D. Zigone et al.

p 3, l 32: If the sensors were repeatedly visited, why did problems with cables persist?

Consider a homogeneous, standard date format.

P 5, l 26: "distortion" of EGF: In the present context, "decrease in signal-to-noise" ratio would perhaps be better.

p. 7, l 5-8: Is there an inversion performed to conclude the 5 m top-layer-thickness? Could the authors briefly comment on the main ingredients?

p. 7, l 10 ff: Is the symmetry larger 15 km/ asymmetry smaller 15 km a persistent feature of all

correlations? Or is it somehow related to just station DF01?

P 7, l 15 ff: Overlapping of pulses at short wavelength is not indicative or related to asymmetry. In general, I find the discussion as to why EGFs are asymmetric at distances smaller than 15 km confusing. How are the envelopes constructed and group velocities estimated? On each lag-side individually? Or are negative and positive lag EGFs stacked?

P 8, l 3: Can error bars be added to Fig 10? They are also given in Fig. 11.

p 9, l 16: I find it too vague and not supported by robust observations to talk about "inside the fault area", if this refers to the low-velocity region that is imaged by the adhoc 2-group approach (see above).

---

## Referee Comment (RC2) · Anonymous Referee #2 · 19 Aug 2016

This paper presents a seismological study of a postglacial fault in Finland instrumented by a local seismic network of 12 seismometers. The array recorded during 20 months. After rejecting the mine blast events, the authors found 40 natural seismic events, with tens of them originating from the postglacial fault. The authors studied the ambient noise recorded by the array. The deduced a 5 m thick quaternary sedimentary layer from H/V ratio analysis. By inverting group velocity dispersion curves extracted from ambient noise cross-correlations, they showed that the seismic velocities in the vicinity of the fault are significantly lower than further away. They concluded that even if the postglacial fault seems non-active from a regional seismic network point of view, a more careful a closer analysis shows that these faults are still active and that they did

not heal since their creation 9000-15000 years ago.

The topic of this study is of great importance to characterize the seismic hazard in a certain areas where hidden or supposedly non-active faults can present a serious threat for the populations. It clearly shows that often, regional seismic networks don't have the sensitivity to detect micro-seismicity evidencing the potential activity of such faults. Some areas that were thought safe may actually be not.

I am a little bit less enthusiastic about the ambient noise analysis, both the H/V analysis and the dispersion curves measurements from the noise cross-correlations needs some clarifications.

I am quite surprised by the high frequency resonance frequency at 30 Hz, which seems very high compared to what is usually found in the literature. Parolai et al. (2002, BSSA) derived an empirical law for the relationship between the depth of the layer and the resonance peak and found that a 5 m layer would resonate between 5 and 10 Hz. However, their shear-wave velocity is different. Can you explain how you found this value of 5 m: is it an inversion, a fit from empirical relationship?

For the dispersion analysis, I suggest to show some Frequency-Time analysis diagrams, so that the reader can see by himself the fundamental mode and the first overtone because they are not obvious from the correlation waveforms shown in the paper.

Also, I understand that the dispersion curves can be noisy and hard to pick, but I would suggest to do a full 2D inversion of the individual dispersion curves to compute group velocity maps of the area covered by the array. These maps can then be inverted at depth to produce a 3D velocity model of the fault zone. The results would be more convincing than the inversion of two ad hoc averaged dispersion curves to show the low velocity around the fault.

These are the main reasons why I would ask for a major revision before publishing this paper.
Specific comments:

- Page 2, Lines 25-28-30: the acronym of 'postglacial fault' should be define at the first occurrence and be consistent all along the text (use always PGF for instance).

- Page 5, lines 3-4: Provide a figure showing examples of the two waveform groups along with their spectra.

- Page 5, line 28: typo → 'f'rom

- Page 6: The description of the beamforming procedure is not clear. Do you perform the beamforming of the cross-correlations or on the raw seismic noise? What do you call 'surface wave parts' (line 15). You should consider to write the beamforming equation you used to make everything clearer. - Page 7, line 13: why don't you use the whitening, it is often necessary to use it in order to obtain reliable correlation functions and dispersion curves. You should at least try both, with and without to see the difference.

- Page 7: The explanation of the asymmetry of the correlation functions is dubious. It is not the distance between the stations that creates this asymmetry, but the noise sources strength azimuthal distribution.

- Page 7, line 24: Specify what type of velocity you are measuring (group or phase).

- Page 8, lines 3-6: The inter-station distances for group 1 pairs may be significantly smaller than for group 2 pairs (and with a main NW-SW orientation). Can the difference of velocity be explained by the difficulty to pick the dispersion curves for short distance station pairs or from a bias due to a predominant direction of noise sources. Using the whitening could also help to 'homogenize' the nose source distribution.

- Page 8, line 15: remove the first 'seismic'

- Page 8: You use the Neighbourhood Algorithm to invert the dispersion curves at depth: what parameters do you invert (how many are they?) and what parameters

boundary do you used?

- Page 9, line 23: typo, 1200 m

- Page 9, line 23: Can this high velocity layer seen at 1200 m for both models be an artifact due to the fact that you set the depth parameter boundary for the last layer around 1200 m, so the inversion cannot find a deeper layer?

- Figure 1: Show an inset of a larger view of the geographical area show in the main figure. You should also consider to use the same coordinate system and coordinate boundary for all the map that you show in the different figures.

- Figure 2: Merge figure 2 and 3 and show the fault on the map.

- Figure 4 and 5: use a logarithmic color scale to better show the details of the spectrograms. And use always the same amplitude limits to help for the comparison between the different panels.

Figure 6: Plot the dates in abscissa instead of the number of days. We we see the figure we believe that the data point are continuous whereas there is a big gap between the dates. It's misleading.

Figure 9: Show the Frequency-Time diagram with the picked dispersion curves on top of it

Figure 10: Show every dispersion curves from both groups along with their respective average.

---

## Author Comment (AC1) · 14 Oct 2016

Dr. Malehmir alireza.malehmir@geo.uu.se

Dear authors: I went quickly through this discussion article and would like to raise a few minor comments: You write: "The length of the PGFs may vary from 2 to 150 km and the maximum height of the fault scarps from 1 to 12 m, yet in the extreme cases up to 30 m (see compilation in Olesen et al., 2004)." The paper by Olesen et al. (2004) likely does not include the most recent PGFs reported by Mikko et al.; Smith et al. and Malehmir et al. from central part of Sweden among others and this could be included

and mentioned that these faults are no longerrinter-friendly version confined to only northern parts of the northern countries as previously thought to. However, they may not be active like others in the north and in your case.

Reply: We are thankful for Dr. Malehmir for his notice about new relevant studies. Publication Mikko et al.; Smith et al. and Malehmir et al added to references list. Corresponding text added to the Introduction part.

You write: "That is why the most plausible explanation is that the low velocity area is a fractured zone inside the fault that can correspond to the fault damage zone (FDZ)." Multi-phase deformation zones are likely responsible for the location of most of PGFs. See the recent study by Malehmir et al. (the same journal) and how they conclude an existing and earlier structure was responsible for the Bollnäs PGF and the complexity of the situation. A major low-velocity zone was also observed there to an extent that delineation of a fresh bedrock movement was impossible to observe (highly fractured and crushed rocks).

Reply: We agree with Dr. Malehmir, but we also would like to notice that the low-velocity zone responsible for Bollnäs PGF and revealed by the high-resolution geophysical study in Malehmir et al. is of smaller scale than the area considered in our study. The former can be attributed rather to the fault core. In our study we obtained integrated characteristic of the area that most probably contains not only the fault core, but also the fractured area around the core. Fault growth commonly produces a fault core composed of slip surfaces and comminuted rock material, and also a broader volume of distributed deformation called the damage zone.

There are also typos in the text here is an example I spotted: The velocity boundary at a depth of about 1200 km is seen in both velocity models obtained from average dispersion curves for Group 1 and Group 2. I guess this meant 1200 m!

Reply: Typos in the text were corrected and references are added to the list and referred to in the text.

Suggested refs: Brandes, C., Winsemann, J., Roskosch, J., Meinsen, J., Tanner, D.C., Frechen, M., Steffen, H., and Wu, P.: Activity of the Osning thrust during the late Weichselian: Ice-sheet and lithosphere interactions, Quaternary Sci. Rev., 38, 49–62, doi:10.1016/j.quascirev.2012.01.021, 2012. Malehmir, A., Andersson, M., Mehta, S., Brodic, B., Munier, R., Place, J., Maries, G., Smith, C., Kamm, J., Bastani, M., Mikko, H., and Lund, B., 2016. Post-glacial reactivation of the Bollnäs fault, central Sweden - a multidisciplinary geophysical investigation. Solid Earth, 7, 509–527. Mikko, H., Smith, C. A, Lund, B., Ask, M., and Munier, R.: LiDAR-derived inventory of 25 post-glacial fault scarps in Sweden, J. Geol. Soc. Sweden, 137, 334–338, doi:10.1080/11035897.2015.1036360, 2015. Smith, C., Sundh, M., and Mikko, H.: Surficial geologic evidence for early Holocenenter-friendly version faulting and seismicity, Int. J. Earth Sci., 103, 1711–1724, 2014.

Reply: The references are added. Best regards, Alireza Malehmir

---

## Author Comment (AC2) · 14 Oct 2016

Dear Editor,

Please consider my comments to the manuscript entitled Structure of Suasselka Postglacial Fault in northern Finland obtained by analysis of local events and ambient seismic noise by Afonin et al. (doi 10.5194/se-2016-90). The authors analyse data from a temporary network consisting of 12 stations deployed in the vicinity of the Suasselka post-glacial fault to study the fault inner structure. Many analysis tools are used, which support a complementary approach to fault zone characterization. However, I feel that parts of the analysis and parts of the presentation and discussion can be improved. The manuscript should be suitable for publication in Solid Earth Discussions

after some moderate to major revision.

I waive my anonymity,

Gregor Hillers

General comments:

The authors discuss aspects of low-velocity damage zones and some resulting implications for faulting. However, in the introduction, they refer only to damage zones in strike-slip faulting environments. These damage zones are persistent features of mature strike-slip fault zones. They evolve over geologic times; their width and the velocity reduction depends roughly on the cumulative fault offset. In addition to this persistent structure, the degree of the velocity reduction, being a proxy for the material's degree of granularity, varies on time scales associated with the seismic cycle. How does this concept relate to the context of post-glacial faults discussed here?

Reply: Fault damage zones have been reported not only for strike-slip faults, but other fault types as well. For example, Kim et al., (2004) in their detailed review demonstrated that damage zones show very similar geometries across a wide range of scales and fault types, including strike-slip, normal and thrust faults. Kim et al. (2004) also demonstrated the general complexity of fault damage zones (in particular, those activated by mutual slip events). Such zones generally are characterized by multiple fracture patterns. That is why we think that the concept of fault damage zone can be used in context of post-glacial faults as well. The correspondent reference to the paper by Kim e al. (2004) is added to the Introduction part.

Comment: What is the cumulative offset of such faults in general and of the Suasselka fault in particular? What is the sense of motion? Under these circumstances, is the signature of a distinct low-velocity zone expected to be resolved?

Reply: Recent studies suggest that the seismic activity in the area of the SPGF continued from 9730 to 5055 call BP (Sutinen et al., 2014). Thus one would expect that this

long-term activity resulted in formation of complex structure of the fault zone. However, the present state of knowledge about the SPGF do not give answer to the question concerning sense of motion and cumulative offset of this particular fault system. Our study revealed some general characteristics of this zone, but further studies would be necessary in order to answer all this questions.

Comment: Considering the earthquake analysis, the database of (downdip) earthquakes observed with the network could be further considered for reconstruction of observables that can be used to infer fault structure. (The authors claim that 2-15 km deep seismicity indicates that the fault is stillactive (line 14 page 5). Does the fault reach 15 km deep?)

Reply: In our study we discovered a number of seismic events recorder by the DAFNE array and originating from the depth of 2-15 km. The depths distribution for events originating from the particular area of the fault zone is shown in the new Fig. 3 (inset). It is seen that the deepest event originate from the depth of 8.5 km. The correspondent explanation of the figure is added to Page 7, lines 13-15.

Comment: Fault zone head or trapped waves can provide important, high-resolution information on material contrasts along and within fault zones. Distinguishing events that do or do not excite head or trapped waves indicate dis-continuous structural elements.

Reply: Unfortunately, in our study it was not possible to analyze trapped waves. The main reason for this was that the events considered in our studies were weak compared to those used for investigation of trapped waves in modern active fault areas (one needs to remember that we are dealing with the area of intraplate seismicity and post-glacial fault). That is why the signal from them was seen clearly at the nearest stations only. At remote stations the signal-to-noise ratio was poor and correlation of phases correspondent to trapped waves was not possible.

Comment: Considering the noise analysis, the authors group cross-fault correlation pairs (termed Group 1) and pairs of stations located on either side of the fault (Group

2). (How are pairs including the 6 stations (50% of the stations) located on top of the mapped fault (Fig. 1) classified?). This approach may certainly be motivated by the relatively sparse station spacing. However, it also reflects the first-order concept of a low-velocity zone sandwiched between two competent blocks. I think the authors must discuss the application of this model better.

Reply: Separation of station pairs into two groups was based on analysis of distribution of all dispersion curves that is bi-modal (see the new Fig. 12). More detailed information about two groups of stations is added to text (P 11, L 3-10).

Comment: Then, the width of the inferred low-velocity region from group-1 results is simply estimated to correspond to the smaller wavelength in the considered frequency band, 1.5 km. This is quite a large value for a fault characterized by a small cumulative offset (the authors may compare this to values obtained from the normal fault hosting the L'Aquila earthquake or any other well-studied fault exhibiting similar characteristics, but not strike slip faults in California).

Reply: We added comparison to the L'Aquila fault zone properties estimated from special distribution of seismicity in the paper by Valoroso et al., 2013 (Page 13, Lines 13-18). They showed that the width of the L'Aquila fault zone varies along strike from 0.3 km where the fault exhibits the simplest geometry and experienced peaks in the slip distribution, up to 1.5 km at the fault tips with an increase in the geometrical complexity. This is in good agreement with the rough estimate of the SPGF zone width in our study.

Comment: What prevents the authors from regionalizing the results as in doi:10.1007/s00024-014-0872-1? First, dispersion curves can be obtained from each pair. This yields, second, 2-D lateral group velocity maps on a grid, which can, third, be inverted for local shear wave profiles.

Reply: Unfortunately, we were not able to find the paper using doi provided by the reviewer (probably there is a typo in doi number). But at the early stage of our research

we tried to calculate 2D velocity sections. Our major conclusion from this exercise was that it would be better to provide reliable first-order approximation of the fault zone area than not very reliable 2D model of the area. For station pairs installed on the same sides of the fault there were too few dispersion curves for reliable 2D results, but scatter and bimodal distribution of dispersion curves is a very well documented feature (see Fig. 12 of the revised manuscript). Therefore, we calculated 2 averaged dispersion curves and solved inversion problems for each of them. These models can be considered as the first-order approximation of the general structure of the fault zone. We hope that our paper would motivate further studies of this particular fault zone with denser network and better spatial resolution.

I think the figures can very much be improved. Is the data from the "areomagnetic" map used as a background synonymous with topography? If not, why is (areo)magnetic data used in the background? Figures 1-3 can be merged into two figures, perhaps even into just one. Please provide a large-scale inset that shows the target region, say, in relation to Scandinavia or Finland. The red "lines" under the DAFNE network in Figure 1 appear as rectangle, boxes. Show the seismicity in Figure 3 in relation to the mapped Suasselka fault. Consider an inset that shows the depth distribution of the seismicity (with error bars).

Reply: We made a new Figure 1 with topography as a background and merged Figures 3 and 4 into one. A subplot with depth distribution of seismicity is also shown in Fig. 3. The error bars differ insignificantly for depth determination, that is why they are not shown. The correspondent explanation is given in the text (Page 6 Lines 9-11). It is known that aeromagnetic maps provide important input for geological mapping. In our study the aeromagnetic map is selected as a background because it shows position of tectonic boundaries and ancient sutures in the region.

Comment: Figure 4: Why not use white for zero amplitude. The figures all look the same; that is what the authors point out in the text, but I wonder whether elimination of redundant data wouldn't be a better strategy for visualization here. I find the color

range not exhausted in Figure 5.

Reply: Figures with spectrograms were corrected (now zero amplitudes are denoted by white colour).

Specific comments: page 1, lines 11, 17: I doubt whether the network with station spacings on the order of 10 km allows probing of the "inner structure" of the fault, except, as said, if high-resolution approaches based on fault zone trapped and head waves are considered.

Reply: Depth distribution of seismicity also provides information about the inner structure of fault zones (c.f. Valoroso et al., 2013). In our study we managed to find events originating from the SPGF fault zone and estimated their depths. As we explained in reply to previous comment, usage of trapped waves was not possible.

p 1, l 28: More than one rupture is needed to create a low-velocity zone (see above).

Reply: In the case of the SPGF it was more than one rupture as the seismic activity in the area of the SPGF continued from 9730 to 5055 call BP (Sutinen et al., 2014).

P 2, l 15: Consider references to earthquake and noise tomographies in fault zone environments

Reply: the proposed references are added.

Allam, A.A., Ben-Zion, Y. & Peng, Z. Seismic Imaging of a Bimaterial Interface Along the Hayward Fault, CA, with Fault Zone Head Waves and Direct P Arrivals Pure Appl. Geophys. (2014) 171: 2993. doi:10.1007/s00024-014-0784-0

Zigone, D., Ben-Zion, Y., Campillo, M., Roux, P., Seismic Tomography of the Southern California Plate Boundary Region from Noise-Based Rayleigh and Love Waves et al. Pure and Applied Geophysics 172,5:1007-1032, 2015 DOI: 10.1007/s00024-014-0872-1.

p 3, l 32: If the sensors were repeatedly visited, why did problems with cables persist?

Consider a homogeneous, standard date format.

Reply: Our stations were installed during September-October, 2011, and all the cables were secured by plastic tubes and buried. In Lapland, the temperatures in winter reach -40 C, and the ground is frozen, so it is absolutely not possible to extract the cables from the ground until June, even if the wrong polarity is seen in the data. Moreover, changing the cables as such remote stations during the experiment is equal to stopping the registration for 1-2 weeks, because the cables could not be repaired directly at sites. As our priority was continuous registration, we solved the problem by making correspondent changes in the final miniSeed data.

P 5, l 26: "distortion" of EGF: In the present context, "decrease in signal-to-noise" ratio would perhaps be better.

Reply: The text was corrected

p. 7, l 5-8: Is there an inversion performed to conclude the 5 m top-layer-thickness? Could theauthors briefly comment on the main ingredients?

Reply: There are no inversion results for top-layer thickness. We estimated this thickness from H/V analysis and used the relationship between thickness of the soft layer, resonance frequency and the existing petrophysical data about seismic velocities in our region (http://www.geopsy.org/documentation/geopsy/hv.html).

p. 7, l 10 ff: Is the symmetry larger 15 km/ asymmetry smaller 15 km a persistent feature of all correlations? Or is it somehow related to just station DF01?

Reply: We observed the asymmetry for all pairs of stations with interstation distances smaller than 15 km (not only for DF01). Figure with EGF calculated between DF01 and each other stations, is presented as an example.

P 7, l 15 ff: Overlapping of pulses at short wavelength is not indicative or related to asymmetry. In general, I find the discussion as to why EGFs are asymmetric at distances smaller than 15 km confusing. How are the envelopes constructed and group

velocities estimated? On each lag-side individually? Or are negative and positive lag EGFs stacked?

Reply: Overlapping of pulses for waves with length more than interstation's distances, of course, is not related with asymmetry, we just observe two parts of EGF together as one part because time shift for casual and acasual parts of EGF's is less or equal to about one period. EGF with overlapping of pulses appears as the asymmetry, therefore in the text we used the term "asymmetry". We changed it to "apparent asymmetry" in the revised version of the manuscript. We estimated group velocity by lag with the largest coefficient of correlation. But if coefficients of correlations are identical for positive and negative lags, we calculated the averaged velocity.

P 8, l 3: Can error bars be added to Fig 10? They are also given in Fig. 11.

Reply: Figure 10 (Fig. 12 of the revised version) is modified and all set of EGF is shown. This gives the information about scatter of two averaged dispersion curves.

p 9, l 16: I find it too vague and not supported by robust observations to talk about "inside the fault area", if this refers to the low-velocity region that is imaged by the adhoc 2-group approach (see above).

Reply: we think that two models obtained in out study can be considered as the first-order approximation of the SPGF area. This first order approximation clearly show the existence of low velocity area associated with the SPGF. But further studies with largest resolution would be necessary to get more detailed information.

---

## Author Comment (AC3) · 14 Oct 2016

Comments by Anonymous Reviewer

This paper presents a seismological study of a postglacial fault in Finland instrumented by a local seismic network of 12 seismometers. The array recorded during 20 months. After rejecting the mine blast events, the authors found 40 natural seismic events, with tens of them originating from the postglacial fault. The authors studied the ambient noise recorded by the array. The deduced a 5 m thick quaternary sedimentary layer from H/V ratio analysis. By inverting group velocity dispersion curves extracted from ambient noise cross-correlations, they showed that the seismic velocities in the vicinity of the fault are significantly lower than further away. They concluded that even if the

postglacial fault seems non-active from a regional seismic network point of view, a more careful a closer analysis shows that these faults are still active and that they did not heal since their creation 9000-15000 years ago. The topic of this study is of great importance to characterize the seismic hazard in a certain areas where hidden or supposedly non-active faults can present a serious threat for the populations. It clearly shows that often, regional seismic networks don't have the sensitivity to detect micro-seismicity evidencing the potential activity of such faults. Some areas that were thought safe may actually be not. I am a little bit less enthusiastic about the ambient noise analysis, both the H/V analysis and the dispersion curves measurements from the noise cross-correlations needs some clarifications. I am quite surprised by the high frequency resonance frequency at 30 Hz, which seems very high compared to what is usually found in the literature. Parolai et al. (2002, BSSA) derived an empirical law for the relationship between the depth of the layer and the resonance peak and found that a 5 m layer would resonate between 5 and 10 Hz. However, their shear-wave velocity is different. Can you explain how you found this value of 5 m: is it an inversion, a fit from empirical relationship?

Reply: In our study we used the Geopsy software (www.geopsy.org) and correspondent recommendation for H/V spectral ratio analysis and interpretation in terms of thickness. It is approximate value. According to petrophysical data, the S-wave velocity estimation for the uppermost sedimentary layer in this region is about 300-500 m/s. Therefore, thickness of this layer is about 3-6 m and the averaged thickness is about 5m.

Comment: For the dispersion analysis, I suggest to show some Frequency-Time analysis diagrams, so that the reader can see by himself the fundamental mode and the first overtone because they are not obvious from the correlation waveforms shown in the paper.

Reply: For dispersion curves estimation we did not use frequency-time analysis, as we noticed that in the same EGF only fundamental or the 1-st higher mode prevails (but not two of them appear together). Moreover, for some station pairs we observed

only fundamental or the 1-st higher mode. Therefore, it was not possible to show them together in one spectrogram. Figure 11 with 2 modes is shown just as an example that two modes were seen in the data.

Comment: Also, I understand that the dispersion curves can be noisy and hard to pick, but I would suggest to do a full 2D inversion of the individual dispersion curves to compute group velocity maps of the area covered by the array. These maps can then be inverted at depth to produce a 3D velocity model of the fault zone. The results would be more convincing than the inversion of two ad hoc averaged dispersion curves to show the low velocity around the fault.

These are the main reasons why I would ask for a major revision before publishing this paper.

Reply: This is the question that was asked also by Reviewer 1. At the early stage of our research we tried to calculate 2D velocity sections. Our major conclusion from this exercise was that it would be better to provide reliable and stable solution for the first-order approximation of the fault zone area than not very reliable 2D model of the area. For station pairs installed on the same sides of the fault there were too few dispersion curves for reliable 2D results, but scatter and bimodal distribution of dispersion curves is a very well documented feature (see Fig. 12 of the revised manuscript). Therefore, we calculated 2 averaged dispersion curves and solved inversion problem for each of them. These models can be considered as the first-order approximation of the general structure of the fault zone. We hope that our paper would motivate further studies of this particular fault zone with denser network and better spatial resolution.

Specific comments: - Page 2, Lines 25-28-30: the acronym of 'postglacial fault' should be define at the first occurrence and be consistent all along the text (use always PGF for instance).

Reply: The first occurrence of the acronym of 'postglacial fault' is defined in the Abstract (Page 1, line 16).

- Page 5, lines 3-4: Provide a figure showing examples of the two waveform groups along with their spectra.

Reply: The figures 4 and 5 with examples of events of two groups (waveforms and spectrograms) are added.

- Page 5, line 28: typo ! 'f'rom Corrected.

- Page 6: The description of the beamforming procedure is not clear. Do you perform the beamforming of the cross-correlations or on the raw seismic noise? What do you call 'surface wave parts' (line 15). You should consider to write the beamforming equation you used to make everything clearer.

Reply: We applied beamforming procedure in the time domain to cross-correlation functions (see P 6, L 18). Surface wave part of EGF has elliptical polarization, so visual analysis of seismograms of ambient noise on screen is possible to apply and to see whether the waveforms corresponding to noise are correlated or not. We used a standard time-domain beamforming procedure (Rost and Thomas, 2002, and Schweizer et al., 2012). This procedure is implemented into the Seismic Handler Motif software: the selected waveforms are marked on seismograms of all stations of the array on screen, then the beam forming in time domain is applied and provides the value of the azimuth.

- Page 7, line 13: why don't you use the whitening, it is often necessary to use it in order to obtain reliable correlation functions and dispersion curves. You should at least try both, with and without to see the difference.

Reply: we tried both in the beginning of our study, but the difference was not significant, in our opinion, that is why we finally decided not to apply whitening.

- Page 7: The explanation of the asymmetry of the correlation functions is dubious. It is not the distance between the stations that creates this asymmetry, but the noise sources strength azimuthal distribution.

Reply: Overlapping of pulses for waves with length of more than interstation's distances, of course, is not related with asymmetry, we just observe two part of EGF overlapping, because time shift for casual and acasual parts of EGF's is less or equal to about one period. Such EGF with overlapping pulses appear as asymmetric ones, therefore in the text we used the term "asymmetry". We corrected it to "apparent asymmetry" in the revised manuscriprt.

- Page 7, line 24: Specify what type of velocity you are measuring (group or phase).

Reply: We measured group velocities for frequency bands with width of 0.125 Hz. But for dispersion curve extractions these velocities approximately may be taken as phase velocities. The explanation is added to the text.

- Page 8, lines 3-6: The inter-station distances for group 1 pairs may be significantly smaller than for group 2 pairs (and with a main NW-SW orientation). Can the difference of velocity be explained by the difficulty to pick the dispersion curves for short distance station pairs or from a bias due to a predominant direction of noise sources. Using the whitening could also help to 'homogenize' the nose source distribution.

Reply: For station pairs from different groups, the distances are approximately equal.

- Page 8, line 15: remove the first 'seismic' Corrected.

- Page 8: You use the Neighbourhood Algorithm to invert the dispersion curves at depth: what parameters do you invert (how many are they?) and what parameters boundary do you used?

Reply: We added a new Table 3 with parameters of the starting velocity model for inversion of dispersion curves. It shows also the boundaries for model parameters.

- Page 9, line 23: typo, 1200 m Corrected

- Page 9, line 23: Can this high velocity layer seen at 1200 m for both models be an artifact due to the fact that you set the depth parameter boundary for the last layer around 1200 m, so the inversion cannot find a deeper layer?

Reply: We added to Table 3 more detailed information about parameters of starting model. As one can see, we used 4-layered model and the range of parameters variations is large both for boundaries depths and velocities. But solutions with minimum misfit were found for 2 and 3-layered model. Inversion cannot find deeper layers, because of frequencies of the signal.

- Figure 1: Show an inset of a larger view of the geographical area show in the main figure. You should also consider to use the same coordinate system and coordinate boundary for all the map that you show in the different figures.

Reply: We made a new Figure 1.

- Figure 2: Merge figure 2 and 3 and show the fault on the map.

Reply: Figures 3 and 4 are merged into one, and position of fault is shown on the map. We decided to keep Figure 2, as it gives more information about seismicity detected previously by regional networks.

- Figure 4 and 5: use a logarithmic color scale to better show the details of the spectrograms. And use always the same amplitude limits to help for the comparison between the different panels.

Reply: Colour scale in Figures 4 and 5 corrected taking into account also comments of Reviewer #1 (new Fig. 6 and 7)

Figure 6: Plot the dates in abscissa instead of the number of days. We see the figure we believe that the data point are continuous whereas there is a big gap between the dates. It's misleading.

Reply: We used numbers of days because of the data gaps, in order to make the figure more compact. But we provided a detailed explanation in the text about correspondence of dates to numbers of days.

Figure 9: Show the Frequency-Time diagram with the picked dispersion curves on top

of it

Reply: For dispersion curve estimation, we did not use frequency-time analysis, as we noticed that in the same EGF either fundamental mode or the 1-st higher mode prevails. Moreover, for some station pairs we observed only fundamental or 1-st higher mode. Therefore, it was not possible to show them together in one spectrograms. The figure shows just examples of fundamental and 1-st higher mode, because we wanted to demonstrate that two of them were present in the data.

Figure 10: Show every dispersion curves from both groups along with their respective average.

Reply: The figure is corrected (new Figure 12).